# Forest carbon sink neutralized by pervasive growth-lifespan trade-offs

R. J. W. Brienen [1✉], L. Caldwell[1], L. Duchesne[2], S. Voelker[3], J. Barichivich [4,5], M. Baliva[6], G. Ceccantini[7], A. Di Filippo [6], S. Helama [8], G. M. Locosselli[7], L. Lopez[9], G. Piovesan [6], J. Schöngart[10], R. Villalba [9] & E. Gloor[1]

Land vegetation is currently taking up large amounts of atmospheric $CO_2$, possibly due to tree growth stimulation. Extant models predict that this growth stimulation will continue to cause a net carbon uptake this century. However, there are indications that increased growth rates may shorten trees' lifespan and thus recent increases in forest carbon stocks may be transient due to lagged increases in mortality. Here we show that growth-lifespan trade-offs are indeed near universal, occurring across almost all species and climates. This trade-off is directly linked to faster growth reducing tree lifespan, and not due to covariance with climate or environment. Thus, current tree growth stimulation will, inevitably, result in a lagged increase in canopy tree mortality, as is indeed widely observed, and eventually neutralise carbon gains due to growth stimulation. Results from a strongly data-based forest simulator confirm these expectations. Extant Earth system model projections of global forest carbon sink persistence are likely too optimistic, increasing the need to curb greenhouse gas emissions.

[1] School of Geography, University of Leeds, Leeds LS2 9JT, UK. [2] Ministère des Forêts, de la Faune et des Parcs, Direction de la recherche forestière, 2700 Einstein Street, Quebec, QC G1P 3W8, Canada. [3] Department of Environmental and Forest Biology, SUNY-ESF, Syracuse, New York, NY 13210, USA. [4] Laboratoire des Sciences du Climat et de l'Environnement, IPSL, CNRS/CEA/UVSQ, 91191 Gif sur Yvette, France. [5] Instituto de Geografía, Pontificia Universidad Católica de Valparaíso, Valparaíso, Chile. [6] Department of Agriculture and Forest Sciences (DAFNE), University of Tuscia, 01100 Viterbo, Via SC de Lellis, Italy. [7] University of São Paulo, Institute of Biosciences, Department of Botany, Rua do Matão, 277, São Paulo, SP 05508-090, Brazil. [8] Natural Resources Institute Finland, Ounasjoentie 6, 96200 Rovaniemi, Finland. [9] Instituto Argentino de Nivología, Glaciología y Ciencias Ambientales (IANIGLA), CONICET-Mendoza, C.C. 330, (5500), Mendoza, Argentina. [10] Instituto Nacional de Pesquisas Da Amazônia (INPA), Coordenação de Dinâmica Ambiental (CODAM), Av. André Araújo 2936, 69067-375 Manaus, Brazil. ✉email: r.brienen@leeds.ac.uk

Over the past 50 years, terrestrial ecosystems have been responsible for the removal of about one third of anthropogenic carbon emissions[1,2]. This net uptake of carbon has been attributed to a combination of afforestation and expansion of secondary forests[3], as well as possible changes in forest dynamics due to nitrogen deposition and increases in atmospheric $CO_2$ and temperature. In particular, increases in $CO_2$ and temperature, in cold regions, have been suggested as the cause of stimulated tree growth, resulting in an imbalance between growth and mortality rates and net uptake of carbon even in primary forests[2,4]. It is thus likely that forests have helped to slow atmospheric $CO_2$ growth rates caused by fossil fuel burning and cement manufacturing, and based on predictions of Earth System Models[5,6] are widely expected to continue to fulfil this role well into the future. However, the degree to which forests will continue to soak up excess atmospheric $CO_2$ depends not only on the growth response of trees to a changing climate and atmospheric composition, but also on changes in mortality rates that ultimately release carbon back to the atmosphere[7–9]. Increases in tree growth due to e.g. $CO_2$, increases in temperature, N deposition, or growing season length, must eventually result in increases in tree mortality[10]. This negative feedback on carbon storage via increased mortality will offset - at least to some extent - the beneficial effects of increased growth on total carbon storage of forests[9,10]. Our current, incomplete knowledge of the universality and causes of the feedback hinders its representation in Earth System Models and thus is an important uncertainty in predictions of future forest carbon uptake in response to global change[7–9,11].

Permanent forest plot monitoring data show widespread mortality increases, which have been proposed to be related to growth increases[4,12]. However, a direct link between growth and mortality trends usually cannot be established using monitoring data alone since detection of this demographic feedback in long-lived trees often exceeds the length of time that inventory data have been collected. Currently the only practical method of assessing the response of lifespans to growth for long-lived organisms like trees is by using annual tree-rings. Across species, tree-ring studies have shown long ago that there is a trade-off between growth and tree lifespan[13,14]. This trade-off has been attributed to a partitioning of allocation in resources to growth versus survival and is a well-known axis of plant strategies ranging from fast-growing pioneer species at one end, and slow-growing longer-lived, shade- or drought-tolerant species at the other end[15,16]. An increasing number of studies have demonstrated that similar trade-offs occur also within species, with faster-growing individuals having shorter lifespans[17–21]. However, these studies focussed mostly on conifers at high elevations[19,21], or only include a limited number of boreal and temperate species[17,18,20]. Some reports even suggest a lack of such trade-offs[22,23]. Thus, evidence for the growth-lifespan trade-off phenomenon, and the extent of its occurrence across biomes and tree taxa, is still incomplete. In addition, insights on the mechanisms by which fast-growing trees tend to die earlier and the magnitude of its effect on forest mortality and the terrestrial carbon sink remains unclear.

Here we use tree ring data to show that trade-offs between early growth and lifespan occur for a large range of species and environments. Faster growth directly reduces a trees′ lifespan and may explain observed increases in tree mortality. Using model simulations, we find that this trade-off has potentially important repercussions for the future carbon sink.

## Results and discussion
### Observations of growth-lifespan trade-offs globally. We here compile and analyse tree-ring datasets including 110 different

species from the tropics to high latitudes to assess the existence of growth-lifespan trade-offs (see "Methods"). We find that taxa with fast early growth rates have short maximum lifespans and vice versa (Fig. 1a), confirming a widely known trade-off between these traits across species[13,15]. Relationships at the tree level, within species, show remarkably similar relationships between early growth and lifespan (Fig. 1b–d). For example, in *Picea mariana* from Quebec (Fig. 1b), fast early growth strongly selects against trees reaching old age, while the oldest ages within this dataset are overwhelmingly from trees that grew slowly when they were young (Supplementary Figure 3). For nearly all of the examined species (74 out of 82) early growth and lifespan were anticorrelated (Fig. 1c, d and Supplementary Fig. 4). On average, tree lifespan decreased exponentially with 23% reduction in lifespan for a 50% early growth increase. The mean strength of the decay constant of the trade-off was similar across different taxa (Gymnosperms vs. Angiosperms) and across climate zones (Boreal, Temperate and Tropical) (Supplementary Fig. 5a–e). Given that our sample includes species from habitats ranging from closed-canopy tropical moist forests to open arctic forests, these remarkable results suggest that trade-offs are not limited to a few specific species or particular habitats. Our analyses also confirm that the observed trade-offs are not a result of biases due to a focus on living tree samples (Supplementary Fig. 5f, g) or the selection of big trees (Supplementary Fig. 6). The finding of consistently longer lifespans for slow-growing trees may seem in direct contradiction to repeated observations of a greater mortality risk for slow-growing trees that are suppressed or have undergone some other form of damage that did not initially kill the trees[22–25]. These apparently contradicting results can be reconciled, however, by differences in the analyses performed and datasets used. We only used rigorously selected species for which we had a very large number of big trees. Thus, our analysis will include with high probability the longest-living trees for a wide range of early growth categories, and permits us to perform 95th quantile regressions to estimate tree lifespans. Unlike previous studies[22–25], our approach is relatively insensitive to early and intermediate stage mortality processes, which are governed by pre-death growth declines. We thus conclude that growth-lifespan trade-offs within species are likely not detectable using most forest inventory data or remote sensing, and yet are a universally occurring and salient phenomenon influencing forest functioning globally.

**Environmental controls of the trade-off.** Several studies have reported that lifespan is strongly correlated with environmental variables like growing season length and air temperature[20]. In line with this, we find negative correlations between lifespan and mean annual temperature for a number of species (Fig. 2a, b and Supplementary Fig. 7). This raises the possibility that trade-offs between growth and lifespan are controlled by external environmental variables, like temperature, and are not due to faster growth directly influencing the likelihood of mortality. We argue, however, that this is not the case with two lines of evidence. First, our in-depth analyses of the dataset for *Picea mariana* from Quebec shows that, while lifespan and growth covary with various environmental variables, including temperature (Fig. 2c, d), soil type, and crown cover (Supplementary Fig. 8), not one single factor fully explains the variation in growth rate and lifespan. Similarly, we observed comparable trade-offs in species growing in ecosystems with intrinsically different growth limitations, including boreal forests which are primarily controlled by temperature (Fig. 2a), tropical floodplains with strong edaphic controls[26], and closed-canopy forests with high levels of competition for light[20,27]. Secondly, growth-lifespan trade-offs remained

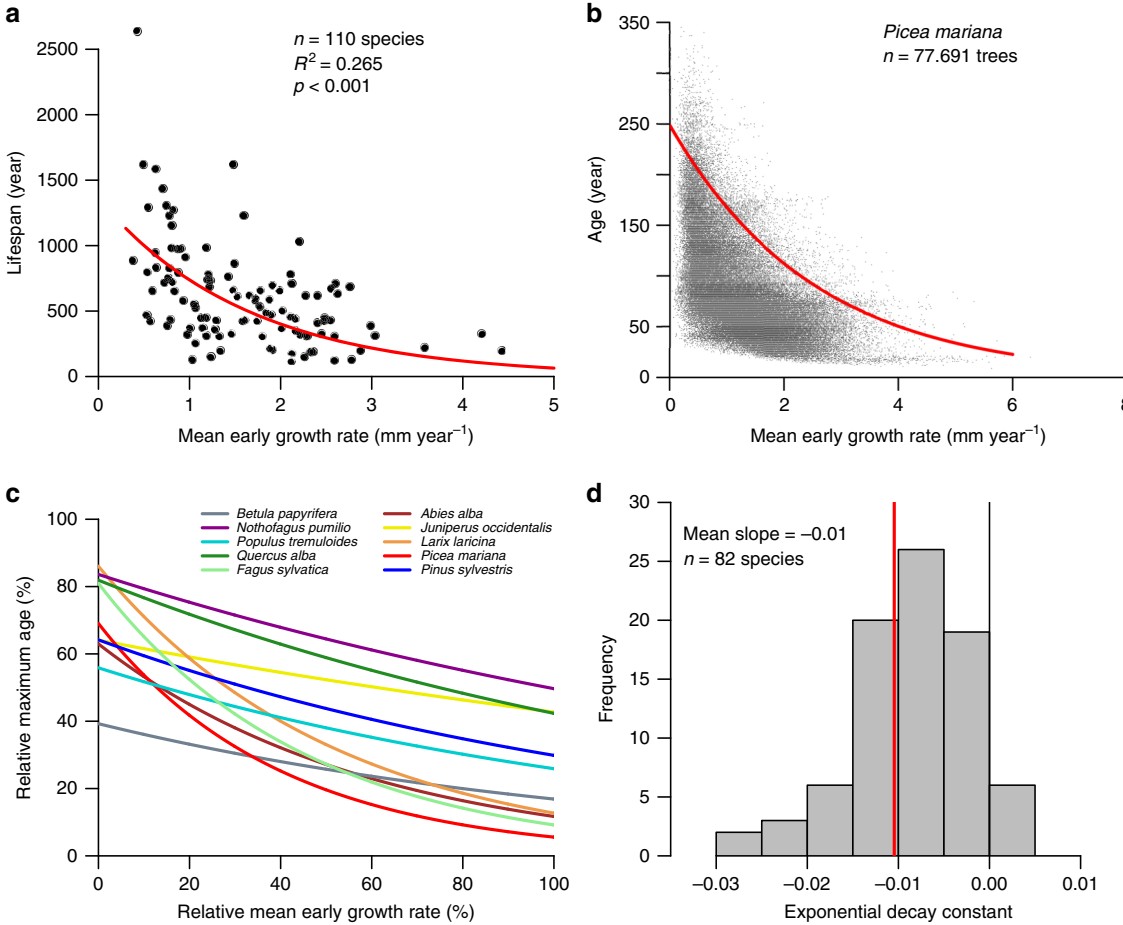

**Fig. 1 Relationship between growth rate and maximum lifespan derived from tree rings. a** Mean early growth rate (mean ring width over first 10 years) versus maximum lifespan for 110 species, and estimated early growth rate-lifespan relationship (red line) using negative exponential major axis regression. **b** Early growth rate versus age for *Picea mariana*, and estimated early growth rate-lifespan relationship (red line) using negative exponential 95th quantile regression. **c** Estimated relative early growth rate-lifespan relationships within species for five angiosperm and gymnosperm species (for individual plots of these species see Supplementary Fig. 4). Relative early growth and relative lifespan were calculated as the ratio of the early growth rate or age of each tree relative to the maximum growth or age for each species. **d** Histogram of the exponential decay constant of relative early growth rate vs. relative lifespan relationships for 82 species with sufficiently large datasets.

strong even after controlling for possible environmental drivers. For example, for *Picea mariana* from Quebec there was no substantial change in growth-lifespan trade-offs when we grouped data by temperature, whereas lifespan versus temperature relationships largely break down when data were grouped by growth rate (Fig. 2c, d). Similar results are obtained for this species for the other controlling environmental variables like crown cover and soil type (Supplementary Fig. 8). For most species, we find very similar patterns with robust effects of early life growth on lifespan, even when controlling for temperature (but not vice versa, see methods). In all, these analyses prove that growth rate is intrinsically linked to tree lifespan, and this explains why trade-offs are in fact found across species of phylogenetically distant taxa and growing in very different environments.

Various theories have been proposed to explain growth-lifespan trade-offs within a species. The first theory is that faster growth requires higher cell metabolism rates which may have directly negative effects on tree lifespans. This theory has also been called the 'rate of living theory' and is known to explain variation in lifespan within clades of animals[28]. It also applies to woody plants at the organ level[29]. However, it has been observed that plant cells do not senesce in a strict sense as animals cells do[30]. Plant meristem cells continuously divide at relatively low rates and show no real deterioration in function, even in very old trees such as bristlecone

pines (~4800 years)[31]. Secondly, trees may face a direct trade-off between allocation to growth versus investment in safer mechanical and hydraulic architecture, and greater investment in defences. Thus, development of traits that increase species survival may result in lower growth rates[13,15,16]. While this has been observed across species, it is unclear to what degree this could explain within-species variation in growth and lifespan. Variation in traits essential to plant survival such as wood density, hydraulic architecture, and resistance to pests and pathogens can be significant, even within species[32], but it still remains to be fully explored how such traits covary with growth and tree longevity. Indeed, evidence from one conifer species indicates survival during a bark beetle outbreak or drought was associated with low growth rates and traits conferring greater bark beetle resistance or greater hydraulic safety, respectively[33,34]. A third theory is that trade-offs between early growth and lifespan arise simply because faster growing trees attain their potential maximum size earlier. Observations show that tree mortality indeed increases as trees grow bigger[35,36], while it has been shown that tree age is a poorer predictor of trees' physiological performance, in comparison to tree size[30]. If this is true then we expect that maximum tree sizes for given climate and edaphic conditions are broadly independent of growth rates. This is indeed what we find for *Picea mariana*, as fast- and slow-growing trees of this species attain approximately the same maximum size

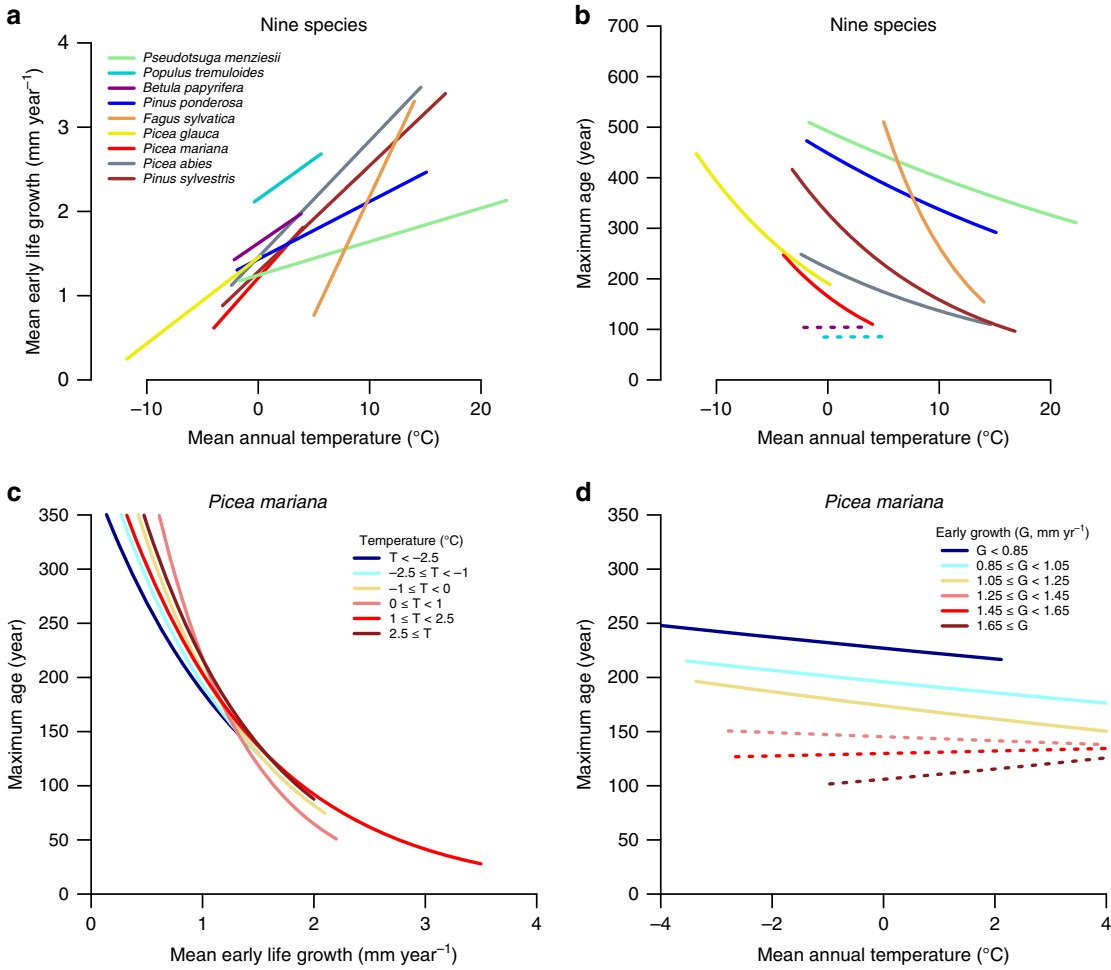

**Fig. 2 Effect of temperature on early growth and tree lifespan. a** Relationship between early growth rates (mean ring width over first 10 years) and temperature for nine different species sampled across North America and Europe. **b** Relationship between maximum lifespan and temperature for the same species. **c** Early growth and lifespan relationships for *Picea mariana* from Quebec for sites within a fixed annual temperature range. **d** Relationship between temperature and lifespan for trees within a fixed early growth rate band. Non-significant relationships ($p > 0.05$) are shown with stippled lines. No adjustments were made for multiple comparisons.

(~300 mm), but do so at notably different ages (Supplementary Fig. 3). As we show in the next section, indeed, simple simulations with observed tree-ring data and mortality rates increasing with size, approaching a potential absolute maximum, result in a very similar growth-lifespan relationship as observed (Fig. 3b). In contrast, age-dependent mortality functions do not reproduce the growth-lifespan trade-off (Supplementary Fig. 9a, b). Thus, the existence of a maximum potential tree size provides a plausible explanation for at least part of the observed trade-off. Maximum tree size is a species-, and possibly site-specific trait[37], but what ultimately kills a tree once it exceeds its maximum potential size may involve hydraulic limitation[38–40], mechanical stability, imbalance between photosynthesis and maintenance respiration, and increasing vulnerability to pathogens and insect outbreaks[41]. While empirical observations of U-shaped mortality size curves[35,36] support this mechanism, we do need to establish the details of every potential mechanism across diverse species to understand the emerging potential for these trade-offs to greatly affect the potential for future carbon storage in forests.

**Implications for forest demography and carbon sink.** We evaluated the effect of observed growth-lifespan trade-offs on forest dynamics using a simple data driven stochastic forest

simulator for *Picea mariana*. Our approach consisted of creating an artificial population by randomly selecting tree-ring trajectories, applying a size-related mortality, and a realistic growth stimulation (Fig. 3c). The applied size-related mortality curve closely matches the estimated size-related mortality rates for *Picea mariana* (Fig. 3a), and results in similar growth-lifespan trade-offs as those observed (Fig. 3b). We then compared the biomass and mortality change over time for simulations that include a trade-off (caused by diameter-dependent mortality), with simulations that do not result in a growth-lifespan trade-off (using age-related mortality rates, Supplementary Fig. 9), and which resemble the approaches commonly used by large-scale vegetation models that predict large biomass increases[5]. Our estimated simulated increase in mean (diameter) growth over 50 years due to northern latitude warming is 29%, roughly consistent with observed temperature driven growth increases of 25% over the past 50 years in boreal western Canadian forests[42] and predicted growth changes at northern latitudes[43].

Our simulations show an initial increase of ~20% in the standing biomass stocks and increases in mortality rates of a similar magnitude. While growth stimulation leads to immediate increases in biomass stocks, mortality starts to increase one or two decades after the initial growth stimulation (Fig. 3). The most important finding of our simulation, however, is that the initial

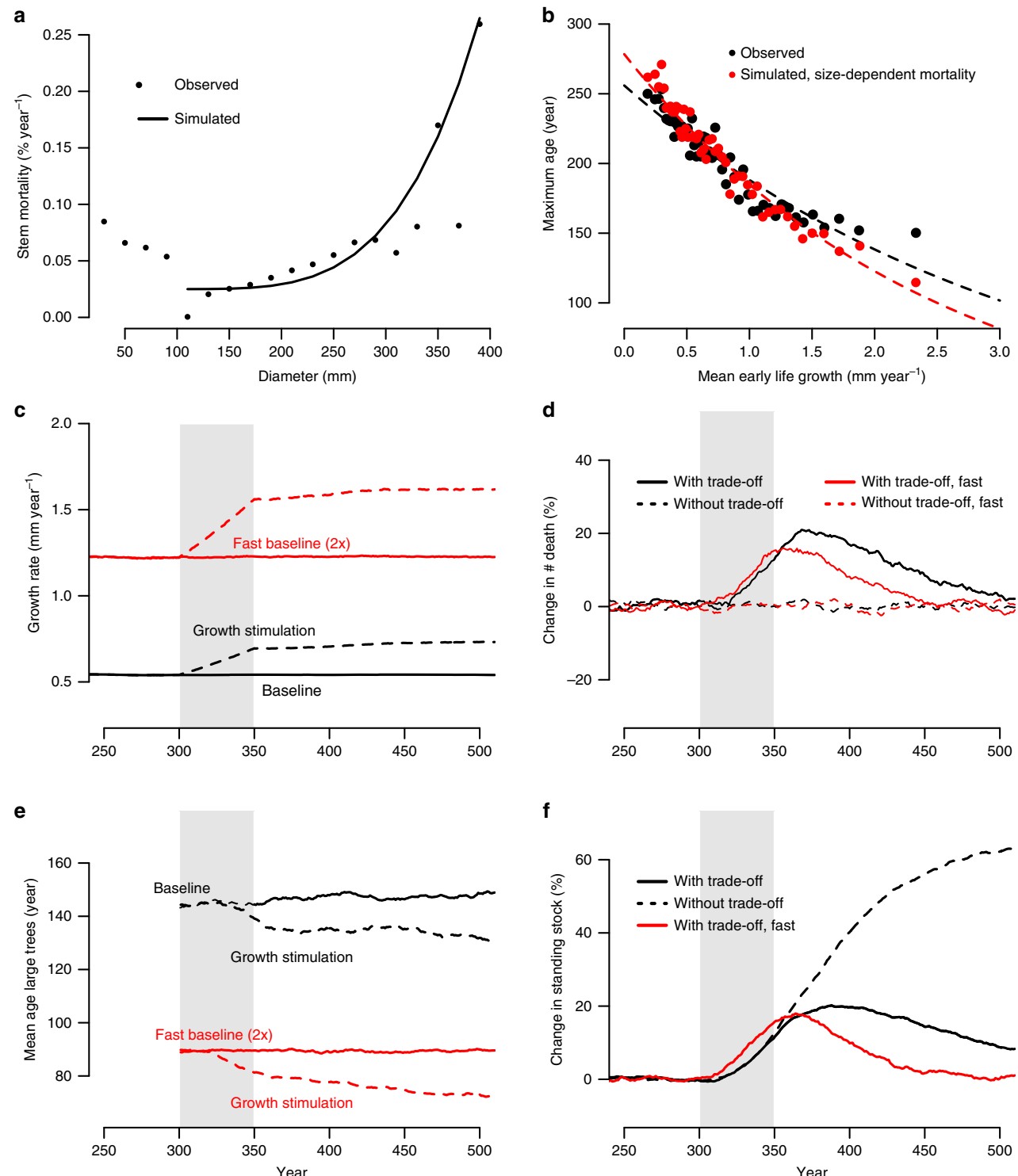

**Fig. 3 Relationships between diameter-mortality and growth rate-lifespan for *Picea mariana*, and simulation results. a** Observed and simulated mortality rates as a function of tree diameter for *Picea mariana* from Quebec, Canada. **b** Observed (black) and simulated (red) relationships between early growth rate and maximum age. **c-f** Effect of tree growth stimulation on mean radial growth rates (**c**), mortality rates (**d**), mean age of large trees at death (**e**), and standing stocks of tree basal area (**f**). Black lines show results for the simulation scenario using the observed tree ring data, and red lines show results for a fast scenario with growth rates two times faster than observed. Broken lines in panels c and e show results for the scenario with growth stimulation, and broken lines in **d** and **f** show results for a scenario based on age-dependent mortality (i.e. without trade-off between growth and tree lifespan). Percent change in tree death rate (**d**) and basal area stocks (**f**) are calculated with respect to the no-growth stimulation or baseline scenario. Shaded area in panels **c-d** indicate the period of simulated growth stimulation from year 300 to 350.

increase in the biomass stocks, the net potential carbon sink, is only transient, and reverses into net biomass losses after the growth stimulation has ceased. Over time the forest biomass stocks revert to the same levels as those observed at the start of the simulation. This progression back toward initial values is entirely due to faster tree growth leading to a reduction of tree lifespans by up to 23 years after growth stimulation ceased (Fig. 3e). In contrast, we find no mortality increases for simulations without realistic growth-lifespan trade-off (i.e., age-dependent mortality, cf. Fig. 3d), and find much higher biomass stock increases which are sustained over time, even after the growth stimulation has ceased (Fig. 3f).

These data-driven simulations suggest that faster growth will result in increases in stem mortality, faster cycling of live biomass, and no true long-term increase in biomass stocks. The simulations rely on several simplified assumptions. Firstly, we did not simulate any competition effects, or changes in tree recruitment. One could argue that changes in climate or $CO_2$ increase will affect recruitment, and increases in standing biomass stocks will increase competition effects, leading to increased self-thinning and even stronger increases in mortality rates. Secondly, we assume that the size-related mortality curve is independent of changes in temperature or $CO_2$, which may not be true. To our knowledge, there is no evidence that potential maximum tree size may increase under higher $CO_2$, whereas increases in leaf-to-air vapour pressure deficits under global warming have been hypothesized to lead to reductions in maximum tree height[44]. It is thus not clear how interacting effects of $CO_2$ and temperature will affect maximum potential tree stature, but it is less likely that maximum tree size will be increased. A more likely scenario that could potentially account for greater future forest carbon storage under rising $CO_2$ is that tree size-density relationships could be modified[45], although long-term empirical data show that the self-thinning rule did not change despite strong growth increases over time[46]. Finally, species distributions are likely to shift in response to climate change, especially in mid to high latitudes, and will affect the total amount of biomass a system can hold[47]. Despite these simplifications, our simulation results are consistent with predictions based on more complex demographic forest models that predict no net biomass increases[9] or strongly reduced increases when including a negative feedback on growth stimulation[7,48]. Our results also bear a strong similarity with some observations of shifting forest dynamics worldwide. Firstly, on-the-ground monitoring studies have shown simultaneous positive trends in growth and mortality rates across the globe[49,50]. Temperature-limited boreal forests experienced growth increases[42] and simultaneous mortality increases[12,51], Central European forests show increases in growth over the past decades leading to accelerated forest dynamics[46], and undisturbed Amazonian forests have experienced long-term productivity enhancements, followed by more recent mortality increases lagging in time by ~20 years[4]. Some of these mortality trends have been attributed to climate variability, in particular changes in the severity and frequency of droughts[4,49,51]. However, here we suggest that mortality increases not only emerge as a direct consequence of increased climate variability, but may also ultimately arise from the pervasive growth-lifespan trade-offs that accelerated the timing of death of large trees.

In summary, we here provide firm evidence for the existence of a universal trade-off between early growth and tree lifespan in trees. Faster growth has a direct and negative effect on tree lifespan, independent of the environmental mechanisms driving growth rate variation. Growth increases, as recently documented across high latitude and tropical forests, are thus expected to reduce tree lifespans and may explain observed increases in tree mortality in these biomes. Data-driven simulations show that

trade-offs have the potential to reduce, or even reverse the global carbon sink of forests in the future. This mechanism is at odds with most extant Earth System Model simulations, which predict a continuation of the carbon sink into mature forests[5], so efforts toward integrating growth rate-mortality trade-offs into process-based simulations of forest carbon storage should receive greater attention.

## Methods

**Tree-ring data**. We used tree-ring records from over 210,000 trees of 110 species, distributed globally in habitats ranging from the tropics to the Arctic region over more than 70,000 sites (Supplementary Fig. 1, Supplementary Table 1). The largest publicly available data source from which we used data is the International Tree-Ring Data Bank (ITRDB, https://www.ncdc.noaa.gov/data-access/paleoclimatology-data/datasets/tree-ring). These were complemented with other datasets to maximize the number of records for each species and to fill in spatial gaps. A particularly large tree-ring dataset used in our analyses is the National Forestry Inventory data from the Ministère des Forêts de la Faune et des Parcs from Quebec, Canada[52,53] (hereafter NFI-Quebec). This data consists of a complete set of ring-width data from 156,711 trees from 79.381 sites across the province of Quebec. Field tree-ring data were collected according to specific standard protocols[52,53], which consisted of selection of up to nine trees in each plot, with 3–5 trees (>91 mm diameter at breast height, DBH) randomly selected, 1–2 selected from the largest trees, and 1–2 from trees closest to the mean tree diameter of the plot[52,53]. From selected trees one core per tree was collected. Tropical tree-ring data were compiled from the ITRDB and from unpublished and published records[26,27,54,55]. For species with larger sample sizes, we distinguished between tree-ring data from trees that died naturally before the moment of sampling and trees that were alive at the moment of sampling, allowing us to test the assumption that living tree ages can be used to estimate trees natural lifespans (see section "Trade-off estimates and assessment of possible artefacts"). Part of these dead tree data were obtained from the ITRDB by selecting tree-ring records of which the last measured ring width was dated to before AD1900. We assumed that most of these trees must have been dead at the time of sampling as no records were collected before 1900. In addition, we compiled published dead tree data from refs. [21,56], and used subfossil tree-ring data from refs. [57,58]. Supplementary Table 1 provides an overview of the datasets, and full details of each dataset are available online as supplemental info.

Various data controls and selection procedures were used to assure high confidence in our dataset. Where possible we tried to identify duplicate records, i.e., multiple cores taken from the same individual tree. This is only a problem for ITRDB, but not for the NFI-Quebec dataset where only one core per tree existed, or for datasets from co-authors. We thus merged ITRDB records that had identical ID's except for the last character of their ID (e.g. 01a, 01b, or ID1-1, ID1-2, etc). From the ITRDB, we only used species for which we could obtain data from a minimum of 3 different sites with at least 20 records each, and only selected species that had a minimum total of 100 separate ring width series. We excluded those sites from the ITRDB that showed relative even age structures, and are thus unlikely to represent old-growth populations that provide robust estimates of trees' maximum lifespans. To this end, we calculated for each ITRDB site the coefficient of variation in tree ages ($CV_{Age} = StandDev_{Age}/ Mean_{Age} \times 100$) and excluded sites with a $CV_{Age}$ lower than 10%. A large subset of ITRDB-data from 46 species has previously been inspected for data quality by co-author S. Voelker[59]. In this subset of data, each ring width series was manually re-aligned by cambial age (i.e., ring number from pith), providing more reliable estimates of tree ages. For the datasets that were not acquired from the ITRDB, we used slightly different criteria. From NFI-Quebec, we used all available sites, excluding those that were classified with evidence of recent management (commercial thinning or clear-cutting) and where fire or insect disturbances destroyed more than 25% of the forest cover.

For estimates of species-level early growth rates and lifespans (cf. Fig. 1a), we included only species with a minimum of 30 records, as lower sample size is unlikely to provide good approximations of tree life spans. This resulted in the inclusion of 110 species, with a median sample size of 305 trees and 12 sites per species. To assess within-species relationships between early growth and tree lifespan, we included 82 species. As a general rule, we included only species with more than a total of 150 trees and from at least 3 sites. About half of our species had more than 300 tree records (see Supplementary Information).

To assess what minimum sample size is needed to get a representative estimate of the true maximum age of a species or a site, and to evaluate how sample size affected estimates of trade-offs between early growth and longevity, we randomly resampled 500 times varying sample sizes—from 25 to 600 trees—from a subset of 11,752 *Picea mariana* trees from NFI-Quebec sites located north of 50.7°N. Comparison of the maximum ages of these random subsets of trees with the true observed maximum ages shows that a sample size of 100 trees results in 99.4% of the cases in maximum age estimates larger than the 95th percentile of the original dataset, and in 67.2% of the cases in ages larger than the 99th percentile original age (Supplementary Fig. 2a, b). As more than 70% of the species had at least 100 trees we thus assumed that for most species, the estimates of their lifespan were close to true lifespans. We used this same approach to assess how sample size affected the estimate of the trade-offs (i.e., estimation of the negative exponential decay

constant; see next section for details). This analysis showed that sample size of 300 trees (corresponding to median sample sizes for trade-off analysis), leads to mean errors in the estimated slope of 12% (Supplementary Fig. 2c). Thus, for most species we achieve relatively accurate estimates of the trade-off strength. Low sample sizes for some species will nevertheless result in small errors of the mean slope, but we expect that positive and negative errors will cancel out against each other. Indeed, we do not observe a specific bias towards over- or under-estimation for low sample sizes, as the mean exponential decay constant for a simulated sample size of 150 trees is very similar to that observed (i.e., −0.409 versus −0.399).

**Trade-off estimates and assessment of possible artefacts.** The strength of the trade-offs between growth and tree lifespan was assessed for each species using a 95$^{\text{th}}$ quantile regression between mean early growth rate and the natural logarithm of age using the QUANTREG package in R[60], as

$$log(A_{(95\text{th quantile})}) = a + b \cdot \overline{\text{RW}}$$

or (1)

$$\text{Lifespan} \approx A_{(95\text{th quantile})} = \exp\left(a + b \cdot \overline{\text{RW}}\right)$$

where $A$ is age of the tree, $\overline{\text{RW}}$ is the mean ring width over the first 10 years. The constant $b$ describes the negative exponential decay constant (i.e., exponential rate of decrease of tree lifespan with increasing early growth rate). This quantile regression fit results in similar estimates of the maximum ages of trees as the 95th percentile ages in binned early growth rate categories (see Supplementary Fig. 3a–c). Note that in contrast the maximum diameter does not vary strongly between slow and fast-growing trees (Supplementary Fig. 3d). We chose a relative short period, the first ten years, for estimating early growth as our study included some relative short-lived species. Previous studies have found similar results when using longer periods (50 years)[56], and we expect no substantial difference using different early growth periods as tree growth is usually strongly auto-correlated in time[61].

To assess trade-off strengths within species, we calculated the mean decay constant ($b$, Eq. 2) for each species using relative age, $A$/max($A$), and relative mean early ring width, $\overline{\text{RW}}$/max($\overline{\text{RW}}$). Maximum of $A$ and $\overline{\text{RW}}$ are species level maxima. The mean slope calculation across all species was weighted by the cube root of the sample size to account for the large differences between species in sample size, and confidence of the trade-off estimates.

While these relationships suggest true trade-offs, they may also be affected (or even driven by) the approaches or analytical methods used here. In particular, we here evaluate the effect of the following four possible artefacts on our results; (1) the use of living trees to estimate tree lifespans, (2) effect of recent growth increases on early growth-age relationship, (3) effects of pith offsets and wood decay on early growth-age relationship, (4) sampling artefacts, such as disproportionate selection of large trees.

(1) Use of living trees: our analysis includes mostly trees that were sampled when still alive, and may thus not be representative for the true lifespan trees may achieve. To assess to which degree use of living trees may affect our results, we analyse and compare the trade-off strengths of trees that died before 1900 to living trees for 12 species with sufficient data availability (minimum of 150 dead and 150 living trees). As the slopes of dead and living trees do not differ significantly (Supplementary Fig. 5f, g, paired t-test exponential decay coefficient, $t = -0.1095$, $p = 0.915$, $n = 12$), we conclude that 95th quantile regressions on living trees can be used to approximate tree lifespan.

(2) Effect of recent growth increases: recent growth stimulation of trees due e.g. to $CO_2$ fertilisation, warming in higher latitudes, and/or nitrogen deposition, may result in observation of a trade-off. This is because recent increases in growth will lead to higher early growth rates for young trees compared to old trees, resulting in a negative relationship between early growth and tree age. The comparison of trade-off strength of dead versus living trees provides strong evidence that this effect does not drive the trade-off. In addition to this, we use a data driven forest simulation (see section "Examining the effects of growth stimulation on forest dynamics") to assess how growth increases affect estimations of the trade-off strength. In this simulation, we used the actual tree-ring data to simulate realistic growth increases of *Picea mariana* tree-ring trajectories in response to high latitude warming. By sampling from these trajectories at the end of the growth increase period (i.e., year 350), and in a period without any recent growth increases (i.e., year 600, see Fig. 3e), we establish that growth increases result in only a small over-estimation of the trade-off, decreasing the exponential decay coefficient from −0.37 to −0.44 (see Supplementary Fig. 9c). Thus, it is unlikely that recent growth stimulation is the cause for the negative relation between early growth and tree lifespan.

(3) Pith offset: tree-ring data, especially those acquired from ITRDB, may miss the innermost sections due to incomplete cores, decayed centres, or imperfect increment borer alignment. Missing rings will result in under-estimation of tree ages and inaccurate early growth rates estimation and could thus affect the estimated relationship between early growth and lifespan. However, ring widths in most species decrease with tree age and size[17], and even trees showing constant wood production with age, will show

decreasing ring width because of geometry. Thus early growth in these samples will underestimate true growth rates and would most likely weaken the observed trade-off, rather than strengthening it. A comparison of species present in both the NFI-Quebec and the ITRDB datasets confirms this. The NFI-Quebec dataset was less affected by pith offset problems, as the trees were carefully screened and trees with substantial differences between cumulative ring widths and field diameters were excluded. Yet, we find that slopes were more negative for NFI-Quebec compared to ITRDB (mean b of −0.25 for Quebec vs. −0.10 for ITRDB, two-sided paired t.test, $t = 2.49$, $p = 0.047$, $n = 7$ species) and pith offsets thus do not explain the relationship. This comparison also shows that estimates of the strength of the trade-offs between early growth and longevity inferred from ITRDB data are probably conservative, as the Quebec data can be considered to be of higher quality, and were collected according to standard protocols. In contrast, data from the ITRDB may contain incomplete series and were collected for unknown purposes, and these issues probably weaken trade-offs in the ITRDB.

(4) Sampling biases: one potential bias in our dataset may arise due to the tendency of tree-ring studies to sample predominantly large trees in the field (i.e., big tree selection bias[62–64]). This may result in a negative relationship between early growth and tree age, as young slow-growing trees tend to be underrepresented in the tree-ring sample (i.e., have not reached the field minimum size threshold yet), compared to fast-growing young trees that are much larger, and therefore more likely to be sampled. This effect would reduce the number of trees with slow early growth and young ages in the tree-ring sample (i.e., trees in the lower left-hand corner of the early growth-lifespan graphs, cf. Supplementary Fig. 6a), and results in overestimation of the 95$^{\text{th}}$ percentile age estimates for slow-growing trees. Our approach to estimate to which degree this bias affected our estimates of growth-lifespan trade-offs was as follows. We first used the tree-ring NFI-Quebec data of *Picea mariana*, combined with plot data from Quebec to reconstruct a new artificial tree-ring dataset with a size frequency distribution identical to the population size frequency distribution for this species in Quebec (Supplementary Fig. 6b). For each tree of *Picea mariana* sampled for their tree-rings we know the early growth rate and age, and also the complete diameter- and age-trajectory up to the year of sampling. From these data, we resampled for each size class (in bin widths of 2 cm) the same number of trees as that observed in the field. By doing this we filled in the lacking data of trees smaller than 91 mm, and created a new artificial tree-ring dataset that had an identical size structure to that observed in the field. We know the mean growth rate over the first ten years and the age at which each individual tree reached the diameter of their respective size class, and could thus reconstruct the early growth rate versus tree age graphs for the full population, including the smaller size classes which were missing from our original tree-ring sample. We then compared the early growth-lifespan relationship for the complete population to that of the trees larger than 91 mm, mimicking the NFI-Quebec field collection protocol. This shows that the exponential decrease is marginally larger ($b = -0.505$ compared to −0.470 for trees >91 mm) and that the intercept is lower (159 years compared to 220 for trees >91 mm) when sampling all trees compared to only trees with diameters >91 mm (Supplementary Fig. 6). Hence, the use of a minimum size threshold (91 mm) in the NFI from Quebec results in a slight underestimation of the trade-off (by ~7%) for the *Picea mariana* dataset. We also resampled from this artificial dataset the 10% largest trees, to mimic a hypothetical standard tree-ring sampling scenario that only samples the largest trees. Such a sampling scenario resulted in a decay constant of −0.432, thus again causing a small underestimation of the true trade-off. This simulation proves that the trade-off is not a result of a sampling bias.

**Possible environmental drivers of the trade-offs.** We evaluated whether the observed trade-off between early growth and tree lifespans could be caused by covariance of growth and lifespan with climate, soil or competition. Temperature variation for example reduces tree growth and lifespan in various species (cf. Fig. 2a, b). To this end, we calculated site-level mean early growth rates and the maximum tree age for a set of species covering different geographic regions (North America, Europe and Quebec). For Quebec, we combined multiple nearby locations to obtain a minimum of 30 trees per site, as sample sizes were low for each location. Site-level mean annual temperature and precipitation was obtained from WorldClim[65]. We then assessed for nine different species the effect of temperature and precipitation on site-level mean early life growth and maximum tree age using major axis regression from the package smart-3[66]. These analyses confirm that early life growth is positively related to temperature for all nine species studied, and that lifespan decreases significantly with temperature for seven out of nine species (see Fig. 2a, b). Using linear mixed effect models with species as random factor (nlme-package-R[67]), we find that across all nine species, mean early life growth increases on average by 0.11 mm for each degree temperature increase, while lifespan decreases by 13 years for each degree temperature increase. Precipitation has no significant effect on early life growth or tree lifespan.

To disentangle whether lifespan decreases are a direct effect of temperature increases, or due to increases in early life growth, mixed effect models were run for all nine species that simultaneously included temperature and mean early life growth rate as explanatory variables for variation in tree lifespan. To account for species differences in growth and age, we used relative mean early ring width ($\overline{RW}$/max($\overline{RW}$)), and relative maximum age ($A$/max($A$)), and used species as a random factor with random intercepts for both early life growth and temperature. This analysis shows that mean early life growth is a stronger predictor of tree lifespans than temperature ($t$-value early growth $= -7.2$, $p < 0.001$, $t$-value temperature $= -2.5$, $p = 0.012$). A similar analysis for *Picea mariana* alone confirms that the primary driver of lifespan is the mean early life growth rate and not temperature, or other environmental variables. For example, analysis of early life growth vs. lifespan in different temperature classes (of 2 °C) remains strong, while the relationship between temperature and lifespan breaks down when this is analysed in growth rate classes (Fig. 2d). Similarly, trade-offs remain strong for *Picea mariana* even when analysing the data in crown cover classes (i.e., an indication of the stand level competition), or when analysing data in different soil classes (Supplementary Fig. 8).

**Examining the effects of growth stimulation on forest dynamics**. We examine the effect of the observed growth longevity trade-off on forest dynamics (growth, mortality, and standing stocks) for a realistic growth stimulation as expected from changes in temperature. We do this by using a data driven forest simulation approach in which we use observed tree-ring data and realistic estimates of size-related mortality rates that result in a trade-off between early growth and lifespan very similar to the observed trade-off. The growth stimulation of trees was estimated using the temperature sensitivity of tree-ring data from the large temperature gradient for Quebec (cf. Supplementary Fig. 7). The full approach is described below.

As demonstrated in our main manuscript, tree-ring data from Quebec reveal a tree growth-longevity trade-off, which may be mediated by rapid increase of mortality with increasing diameter. Not only tree cores, but also detailed forest census data exist for this region, allowing reconstruction of size-dependent mortality relationships. This analysis was done using only data from Quebec forest inventories north of 50.7°N. First, we estimate mortality, μ, as a function of tree diameter using both forest census data inventories and individual tree-ring records from *Picea mariana* from NFI-Quebec. We assume a stationary state of the tree diameter ($D$) number distribution, $N(D)$. Thus, for a diameter class with $D$ in the interval [$D,D + \delta D$]

$$0 = \frac{dN(D)}{dt} = I(D) - L(D) - \mu \cdot (D). \quad (2)$$

Here $I(D)$ is the number of individuals growing into diameter class [$D,D+\delta D$] per unit time, and likewise $L(D)$ the number of individuals leaving the same diameter class per unit time. Thus, mortality can be estimated from the steady diameter distribution, obtained from plot data (Supplementary Fig. 7b), and number of in-growers and leavers, estimated from the diameter distribution and ensembles of randomly selected growth trajectories, as

$$\mu(D) = \frac{I(D) - L(D)}{N(D)}. \quad (3)$$

The result is shown in Fig. 3a, together with a fourth order polynomial model of the form

$$\mu(D) = \begin{cases} 0 & for\ D \leq D_0 \\ b + k \cdot (D - D_0)^4 & for\ D > D_0 \end{cases}. \quad (4)$$

Here $b = 0.025$, $k = 3 \times 10^{-11}$ and $D_0 = 91$ mm, which is the minimum sampling diameter used for the NFI-Quebec. The rationale to set the mortality below $D_0$ to zero in this model is that the trees sampled for tree rings did all survive to this diameter (as only trees with diameters >91 mm were cored). Thus, only by setting mortality to zero for trees <91 mm allows for a proper comparison of simulation results with observed data. We ended up using this model as it reproduced the observed trade-off between growth and longevity quite well (see Fig. 3b). The strength of the simulated trade-off is robust with regard to the choice of mortality function as linearly increasing mortality rates equally reproduced the observed trade-off. Increasing mortality rates towards larger size classes is consistent with observations of size-dependent mortality in temperate and tropical trees[35,36,40,68], and supports the notion of a species-specific maximum size threshold[64]. For *Picea mariana*, the 99th percentile maximum tree size is 353 mm, close to diameters at which we find large increases in tree mortality (cf. Fig. 3a). For comparison purposes, we also calculated an age-dependent mortality rate and performed alternative simulations using this mortality model. The age-dependent mortality relationship was derived in a similar manner to the size mortality curve by repeating the above procedure and calculations (Eqs. 2 and 3) for (yearly) age classes. Age-dependent mortality was parametrized as:

$$\mu(A) = \begin{cases} 0 & for\ A \leq A_0 \\ a + b^* A & for\ A > A_0 \end{cases}, \quad (5)$$

with $a = 0.021$ and $b = 0.0000015$. Analogous to the diameter-dependent mortality

model, we set mortality to zero for trees younger than 74 years ($A_0$), which is the age at which the average *Picea mariana* tree reaches 91 mm in diameter ($D_0$).

We next created a 600 year sequence of annually seeded, 1250 member, tree cohorts. Each member of a cohort is a randomly selected tree diameter growth trajectory derived from the tree-ring cores, extended in time to an age of 500 years. Short growth trajectories were extended by using the mean growth of the 10 oldest trees that had a similar ring width in the first ten years of growth. To this end all early mean ring width was grouped into six equal early ring width classes.

All trees of this so-constructed tree cohort set live, by construction, exactly 600 years. Realistic age structures were realised by sequentially (year-on-year and tree-by-tree) assigning death to trees where a random number generator identified those individuals smaller than μ(D) or μ(A). To test the realism of this procedure, we compared the predicted and observed tree age versus early ring width relationship —or i.e. the growth-longevity trade-off. The slope of the relationship for the diameter-dependent mortality model μ(D) is very close to observed (Supplementary Fig. 9a), and is thus a realistic representation of the observed mortality process and justifies its use to examine the effect of a growth stimulation on standing stocks. In contrast, we find that the age-dependent mortality model μ(A) does not result in a significant trade-off between early growth and tree lifespan (Supplementary Fig. 9b), providing an ideal comparison for models that fail to incorporate the observed trade-offs.

To mimic growth stimulation, we boosted growth of trajectories from year $t_0 = 300$ year onwards of the 600 year sequence of cohorts, while exposing the trajectories over the entire 600 year period to the mortality algorithm just described. We stimulated growth rate, $\overline{RW}$ (mm year$^{-1}$) from year $t_0 = 300$ year onwards according to

$$\overline{RW}_{stim}(t) = \overline{RW}(t) \cdot \exp(\lambda(A) \cdot \delta T(t)), \quad (6)$$

and

$$\delta T(t) = \begin{cases} \frac{dT}{dt} \cdot (t - t_0), & t - t_0 < \tau \\ const, & t - t_0 \geq \tau \end{cases} \quad (7)$$

where $\delta T(t)$ is a normalized temperature trend (year$^{-1}$), $\lambda(A)$ a unitless function of tree age representing that growth sensitivity of young and old trees to temperature may vary with tree age[59], and $\tau$ is 50 years, the duration of the period of growth stimulus. We calculated the observed temperature trend for Quebec over the past 100 years from CRUTEMP data[69], and simulated growth increases from year 300 to year 350 in response to observed warming rate, estimated to be 0.0221 °C year$^{-1}$. We used a space-for-time substitution approach on the full dataset of Quebec[43] to estimate the ring width response of *Picea mariana* to temperature. We conducted these simulations in age-bands of 10 years (0–10, 10–20, …. 140–150, >150 yrs) as younger trees are more sensitive to temperature increases than older trees (Supplementary Fig. 9d), and use an exponential model of the form

$$\overline{RW}(A, T) = a \cdot \exp(\lambda(A) \cdot \delta T), \quad (8)$$

where $a$ is a constant, $\lambda(A)$ the exponential increase rate for age class [$A, A+\delta A$] and $\delta T = (T - 0)$ (°C). We then used the exponential growth increase with temperature for each age band, $\lambda(A)$, to estimate the relationship between tree age and $\lambda(A)$ (Supplementary Fig. 9e). In all cases the age modulation of the stimulus is

$$\lambda(A) = \begin{cases} 0.0000132 \cdot A^2 - 0.00291 \cdot A + 0.229, & A < 135 \\ 0.07, & A \geq 135 \end{cases}. \quad (9)$$

Finally, we compared the effect of growth stimulation on forests dynamics against simulation without growth increases (baseline) for the two different mortality models. We also performed one simulation where we multiplied the full growth series by 2, as a representation of the effect of growth stimulation on a faster-growing species. Finally, we evaluated for each simulation the mean ring width growth, the stem mortality rate, the age of the largest (75th percentile) trees that died and the change in the total basal area stock over time for the full population. For the stem mortality and the basal area stocks, we calculate and present the change in dynamics of the growth stimulation scenario relative to the baseline scenario without growth stimulation. All analyses and simulations were performed using R-studio, version 0.99.903[70]. Maps in SI figures were produced using the ggmap function from the R-package 'ggplot2'[71]

## Data availability

All metadata and early growth and tree age data are available from https://doi.org/10.6084/m9.figshare.12620414.

## Code availability

Source code to reproduce Fig. 1 is available from https://doi.org/10.6084/m9.figshare.12620414. Additional codes used in this analysis are available from corresponding author upon reasonable request.

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

## Acknowledgements

We acknowledge the contributors to International Tree-Ring Data Bank for making available raw tree-ring data that were used in this study, and thank the staff at the Direction des Inventaires Forestiers of the Ministère des Forêts, de la Faune et des Parcs du Québec for sharing tree-ring and sample plot data from the forest inventory program in Quebec, Canada. R.J.W.B. was supported by NERC grant NE/S008659/1, E.G. was supported by NERC grant NE/N012542/1, and G.L. and G.C. were supported by FAPESP grants 12/50457-4 and 17/5008-3, J.B. was supported by the Centre National de la Recherche Scientifique (CNRS) of France through the presidential program Make Our Planet Great Again. S.H. was supported by the Academy of Finland, and A.D.F. was supported by National Geographic Society grant WW-136R-17. L.L. and R.V. were partially funded by CONICET.

## Author contributions

R.J.W.B, E.G. and L.C. designed this study, R.J.W.B. and E.G. wrote the paper, R.J.W.B., L.C. and E.G. carried out data controls, analysis and simulations, J.B. and S.V. contributed to ITRDB data downloads and controls, L.D., S.V., M.B., G.C., A.D., S.H., G.L., L.L., G.P., J.S., R.V. contributed data, and all authors commented on the paper.

## Competing interests

The authors declare no competing interests.
