## [Peer Review File · Nature Communications]

Reviewers' comments first round –

Reviewer #1 (Remarks to the Author):

Brienen et al for Nature Communications

The authors explore what I consider the central issue in projections of global biomass carbon stocking: tree dynamics. This paper is going to have a huge impact, because it will likely put an end to the popular assumption that growth is a surrogate for C storage. Current models either ignore carbon residence time, or they assume mean residence time of C either to stay constant or to increase with rate of growth, and most experimentalists exploring CO₂ enrichment effects can not provide such life cycle responses and thus howl with the crowd by selling their growth rate responses as a stock change (C sequestration). Dendrosciences offer the needed data to fill this gap, and this is what these authors are doing. The evidence provided here is overwhelming, namely that faster growth reduces residence time of C in forests.

I found this text extremely well written, exceptionally well supported by data, and exploring the issue at great depth, including potential mechanisms and a modelling attempt. Thus, I strongly support going ahead with this paper. It needs only a very minor revision in my view. The message is clear-cut and of top importance.

The authors may wish to adjust phrasing at some places (see my Adobe annotated pdf). Most of my comments relate to the introduction. I found the reference to CO₂ enrichment effects on forests a bit light footed, because these effects were all modelled and not measured in forests that had reached a steady state leaf and fine root turnover and a stable nutrient cycle. Those handful experiments that explored CO₂ effects in steady state forest systems found zero effects of elevated CO₂, presumably for a dominance of element stoichiometry and thus mineral nutrient limitation (not just N). Whatever one assumes in this respect, has no what-so-ever effect on the current analysis by Brienen et al. No matter what might stimulate tree growth, the rules distilled here need to become accounted for. In my view, there is no need to perpetuate some - what I consider out-dated, though main stream - reasoning regarding CO₂ effects. A minor issue, indeed.

If, for editorial reasons, the text would be required to be shortened, I see a bit of leeway in the mechanisms part (these points could be packed into more compact statements) and in the modelling part. The model part represents a welcome added value, but the stronghold of this article is the data base and its elegant analysis.

See my comments on the pdf

Christian Körner

Reviewer #2 (Remarks to the Author):

In this manuscript, the authors present evidence of a universal trade-off between tree growth rates and longevity, both within a broadly distributed species (Black spruce) and among species worldwide. Though this notion has been proposed before, it is a compelling idea because of recent work suggesting that tall trees die preferentially in droughts (Stovall et al., 2019), because the causes of tree death remain elusive (Sevanto et al., 2014), and because there is no satisfying evidence of trees' ageing—they just seem to grow larger (Mencuccini et al., 2005). The current manuscript thus tests an idea that sits astride several interesting questions. The authors go on to interpret their data in light of a fourth question: whether shorter maximum lifespans will reduce the accumulation of carbon in forests, which could neutralize their value as a carbon sink in the long term.

As an ecophysiologicalist, I came to this manuscript hoping to find a mechanism. I was unsatisfied by

the correlations with environmental variables that were considered in the analysis and am not convinced they have excluded all possible causes except early growth rate. But that's OK, they are very clear about their assumptions and perhaps someone will find the physiological mechanism later. It's an interesting result anyway.

The data show the trade-off clearly. The question is why are there so few old trees that started out quickly in these tree-ring datasets?

I wonder about the possibility of three potential sources of bias against sampling fast-growing old trees--which would be big trees. First, these big old trees are valuable and they would likely have been logged, leaving the smaller old trees, often in reserves and national parks, to be sampled. Perhaps the Canadian Forest Inventory gets around this issue, but it is not clear from what is written. I presume the inventory was designed to sample some clearly identified population and that population should be specified. Second, because tree-ring samples are often collected in extreme environments to support climate reconstructions, the fast-growing trees in better environments might be underrepresented in the tree-ring dataset. It is not clear to me how much of the dataset comes from the forest inventories and how much uses the existing tree-ring studies. Third, large old trees are often decayed in the center, making the determination of age impossible. I presume that such trees were eliminated from the current study, as they would be impossible to age, but we are not told. It is conceivable that old trees that grew fast in their youth are more likely to be rotten in the center because the conditions for decay organisms are also improved on better sites. Besides, the rapidly produced wood at the center might be less well defended than when it is produced more slowly. If they were more prone to rot, that would drop them out of this sample and contribute to the shape of the data in Fig. 1a and 1b. I ask that the authors tell us how the rotten cores were treated in the sampling and how that might influence the results.

I was interested in the way they have estimated mortality due to increasing growth rates, and am particularly curious about how well they have described self-thinning. It has been noted that self-thinning causes mortality to increase as growth rates have increased (Pretzsch et al., 2014), but that stands still follow the old self-thinning lines. I would like to see how well the new mortality calculations reproduce the self-thinning line. A demonstration of similar patterns would make the analysis more convincing. I understand that they have described maximum ages of survivors, which might be considered independent of self-thinning—except that the growth of survivors was influenced by the self-thinning during their lives—including perhaps their early lives. In any case, I think they should explain this more clearly. And while they're at it, I would like to see some mention of which parameter is most valuable for land surface modeling: average longevity or maximum?

Exceptions to the size-longevity tradeoff might also be worth considering. I would start with the redwoods of northern California and the ancient Douglas-fir forests of west coast of North America (Waring & Franklin, 1979) . Thanks to the late arrival of industrial forestry and the efforts of conservationists, some of these commercially valuable forests were preserved before they could be logged. Here one finds many of the largest, and perhaps oldest, individuals of several conifer genera in the world. I doubt that these individuals grew slowly when they were young, but it would be especially interesting to consider this case, at the other end of the size and growth-rate continuum from black spruce. There must be some such trees in British Columbia in the Canadian Forest Inventory. Perhaps they could be extracted from the mass of data to see if they fit the same patterns.

The conclusion about neutralizing the carbon sink ignores forest management, which captures much of the forest production occurring in some parts of the world(Luyssaert et al., 2018). Sustainable forest harvesting could be accelerated if faster growth occurs, capturing the mortality before the mortality occurs. This would, of course, prevent the forests from reaching the age of natural mortality and would lead to a different set of fates for the carbon, including storage in wood products. In fact, this seems like a logical response to the patterns described in the current manuscript. I do not think this is a quibble if the authors wish to improve the parameterization of land surface models--or if they are interested in reaching a broad readership.

Specific comments (overlapping with the above):

Line 77: Do the global models need the maximum lifespans or the averages? Almost all of tree mortality occurs due to self-thinning or forest management. How relevant are these maxima globally?

Lines 106-131: These are not so much causes as correlations, by my reckoning. As an ecophysiologicalist, I would like to know what physiological process is disrupted uniformly across this range of sites. This is analogous to the cavitation or carbon starvation question...or maybe it's exactly the same question. In any case, this manuscript would be strengthened by a tie to the ongoing development of mechanistic thinking about tree mortality. Should cite Koch et al. about the physiological basis of max ht (Koch et al., 2004) and the influence of height on drought-induced mortality (Stovall et al., 2019)

149-168: tree size height and diameter are intermingled in a way that is confusing here because height limitations might occur due to hydraulics, but allocation to diameter growth might well be favored by height limitations. So there's a question underneath this discussion about size related to allometric shifts with height.

210: But Pretzsch (Pretzsch et al., 2014) argues that so far they haven't. Should say so.

210-212: Right, but there will be significant mortality while the shifts are taking place. Is that within the time frame of these models or are they steady-state?

Fig. 2 errors? Data?

Fig 3a. two data points support the exponential increase at the end, but more do not. In fact, it is not clear to me that the exponential curve fits better than a straight line would. This could lead to a strong overestimate of mortality. It is stated in the text (line 439) that there was no bias in the exponential fits, but this figure seems to show otherwise.

Why ring widths and not basal area increase? The same amount of wood produces a narrower ring on a bigger tree. Does that matter here?

473-480: good

547: No. It shows that the biases you tested were not significant. See questions about other biases above.

Koch, G. W., Sillett, S. C., Jennings, G. M., & Davis, S. D. (2004). The limits to tree height. *Nature*, 428(6985), 851–854.

Luyssaert, S., Marie, G., Valade, A., Chen, Y.-Y., Njakou Djomo, S., Ryder, J., Otto, J., Naudts, K., Lansø, A. S., Ghattas, J., & McGrath, M. J. (2018). Trade-offs in using European forests to meet climate objectives. *Nature*, 562(7726), 259–262. <https://doi.org/10.1038/s41586-018-0577-1>

Sevanto, S., McDowell, N. G., Dickman, L. T., Pangle, R., & Pockman, W. T. (2014). How do trees die? A test of the hydraulic failure and carbon starvation hypotheses. *Plant, Cell & Environment*, 37(1), 153–161. <https://doi.org/10.1111/pce.12141>

Stovall, A. E. L., Shugart, H., & Yang, X. (2019). Tree height explains mortality risk during an intense drought. *Nature Communications*, 10(1), 4385. <https://doi.org/10.1038/s41467-019-12380-6>

Waring, R. H., & Franklin, J. F. (1979). Evergreen Coniferous Forests of the Pacific Northwest. *Science*, 204(4400), 1380–1386. <https://doi.org/10.1126/science.204.4400.1380>

Reviewer #3 (Remarks to the Author):

Overall the paper presents convincing evidence of early growth rate -lifespan trade-offs globally, as well as possible explanations for these trade-offs.

The authors apply simple simulations that assume mortality rates that increase with size. These produce growth-lifespan relationships similar to those that are observed, whereas an assumption of age-dependent mortality does not. The authors conclude that growth-lifespan tradeoffs are missing from current Earth system models which must therefore be incorrectly predicting a continuation of the carbon sink into mature forests into the future.

My main concern with this paper is that the extrapolation of the findings to conclusions about the incorrectness of Earth system models is not well-founded. The reference (3) (Sitch et al. 2008) to models examines projections of Dynamic Global Vegetation Models (DGVMs) not Earth system models. and there has been much subsequent development in the DGVM community. Many DGVMs now include stand demography (including recruitment, competition and size-dependent mortality), from which may emerge growth-lifespan trade-offs. See Fisher et al. 2017, 10.1111/gcb.13910 for a more up to date and comprehensive review of the current generation of DGVMs.

My second main concern is about the final recommendation (l 234) that growth-rate mortality trade-offs be integrated into process-based simulations of forest carbon storage. This is not an immediately helpful suggestion, as the process or processes from which these trade-offs emerge are not well understood- as discussed in the section headed "Causes of the trade-off". A more constructive use of the findings could be to recommend how the data relating growth rate to life-span, or a synthesis thereof, could be used as a benchmark for DGVMs.

Minor comments:

Line 36: this comment on current Earth system models is not well supported by the article.

Line 41: generality here is misleading: increases in temperature don't always lead to growth stimulation.... In warm climates, sensitivity of growth to T may be negative.

Line 45: reference (3) is insufficient to support this: the models assessed by Sitch et al. are not the same as in Earth system models.

Response to reviewers' comments.

Red = reviewers' comment, Blue = our response.

Reviewer #1 (Remarks to the Author):

Most of the reviewer's comments were very helpful and interesting and we thank the reviewer for his efforts on improving this manuscript.

The authors may wish to adjust phrasing at some places (see my Adobe annotated pdf). Most of my comments relate to the introduction. I found the reference to CO₂ enrichment effects on forests a bit light footed, because these effects were all modelled and not measured in forests that had reached a steady state leaf and fine root turnover and a stable nutrient cycle. Those handful experiments that explored CO₂ effects in steady state forest systems found zero effects of elevated CO₂, presumably for a dominance of element stoichiometry and thus mineral nutrient limitation (not just N). Whatever one assumes in this respect, has no what-so-ever effect on the current analysis by Brienen et al. No matter what might stimulate tree growth, the rules distilled here need to become accounted for. In my view, there is no need to perpetuate some - what I consider out-dated, though main stream - reasoning regarding CO₂ effects. A minor issue, indeed.

We understand the reviewer's comment and have tried to scale down the link between CO₂ and growth stimulation. In some places, we nonetheless kept our original text linking growth increases to CO₂, as this is one of the reasons why Earth System Models predict an ever-increasing land carbon uptake. We also maintained to some extent the sink terminology as it is deeply ingrained in the carbon cycle community and otherwise the importance of the article may not be understood, although we sympathise with the reviewer's dislike of this term.

We took over most of the suggestions from the annotated PDF, and list below our response to some of the more important ones.

- In the abstract there was some unclarity regarding the lagged tree mortality, but we could not find an alternative short formulation that conveys the same message and thus have not changed this formulation. We are open to any suggestions however.
- We have changed wording in the introduction to acknowledge the possible large contribution of land carbon uptake due to forest regrowth, and added in the suggested reference of Pugh et al.2017.
- There was a comment on potential confusion of fluxes and stocks, but in our view the formulation as we originally had, is correct. We say "*This negative feedback on carbon storage via increased mortality will offset -at least to some extent- the beneficial effects of increased growth on total carbon storage of forests^{6,7}.*" In our view it is possible that productivity is increasing while mortality is not immediately increasing, thus leading temporarily to an increase in stocks- as we observe (Fig 3). This would not be true if mortality increases at the same time and rate as productivity.
- To reflect the reviewer's comment on the role of plant evolution imposing size limits on specific species, we rephrased the sentence slightly, and now say "**Maximum tree size is a species-specific trait (Thomas, 1996)**, but what ultimately kills a tree once it exceeds its maximum potential size may involve hydraulic limitation³⁸⁻⁴⁰, mechanical stability, imbalance between photosynthesis and maintenance respiration, and increasing vulnerability to pathogens and insect outbreaks⁴¹."

References added:

Thomas, S. C. Asymptotic height as a predictor of growth and allometric characteristics in Malaysian rain forest trees *Am. J. Bot.* **83**, 1570-1570 (1996).

Pugh, T. A. *et al.* Role of forest regrowth in global carbon sink dynamics. *Proceedings of the National Academy of Sciences* **116**, 4382-4387 (2019).

If, for editorial reasons, the text would be required to be shortened, I see a bit of leeway in the mechanisms part (these points could be packed into more compact statements) and in the modelling part. The model part represents a welcome added value, but the stronghold of this article is the data base and its elegant analysis.

We have not shortened the manuscript at this stage but if needed we will come back to the reviewer's suggestion.

Reviewer #2 (Remarks to the Author):

In this manuscript, the authors present evidence of a universal trade-off between tree growth rates and longevity, both within a broadly distributed species (Black spruce) and among species worldwide. Though this notion has been proposed before, it is a compelling idea because of recent work suggesting that tall trees die preferentially in droughts (Stovall et al., 2019), because the causes of tree death remain elusive (Sevanto et al., 2014), and because there is no satisfying evidence of trees' ageing—they just seem to grow larger (Mencuccini et al., 2005). The current manuscript thus tests an idea that sits astride several interesting questions. The authors go on to interpret their data in light of a fourth question: whether shorter maximum lifespans will reduce the accumulation of carbon in forests, which could neutralize their value as a carbon sink in the long term.

As an ecophysiologicalist, I came to this manuscript hoping to find a mechanism. I was unsatisfied by the correlations with environmental variables that were considered in the analysis and am not convinced they have excluded all possible causes except early growth rate. But that's OK, they are very clear about their assumptions and perhaps someone will find the physiological mechanism later. It's an interesting result anyway.

The data show the trade-off clearly. The question is why are there so few old trees that started out quickly in these tree-ring datasets?

I wonder about the possibility of three potential sources of bias against sampling fast-growing old trees—which would be big trees. **First, these big old trees are valuable and they would likely have been logged, leaving the smaller old trees, often in reserves and national parks, to be sampled. Perhaps the Canadian Forest Inventory gets around this issue, but it is not clear from what is written. I presume the inventory was designed to sample some clearly identified population and that population should be specified.**

The vast majority of the sampled territory in Quebec -where we find the strongest trade-offs among all species studied- is pristine forest that has never been harvested historically. In those sites, where historical logging did take place, harvesting consisted of clearcutting, which is the only timber harvesting practice used in boreal forest stands of Québec. This logging practice removes all trees in a stand regardless of their size, and is less likely to cause bias as compared to selective logging which removes only the largest trees within a population. Moreover, logging never took place in Quebec above 50°N and even in central Quebec territory only ca. 30% of total area has ever been logged in the past 50 years. However, most importantly, the forest inventory data classified any partial disturbances for each site, and any forest stands with evidence of recent partial disturbance (harvesting, fire, insect defoliation) where more than 25% of the forest canopy was removed, were

excluded from our analysis. Thus, one would expect that our sample should include at least some fast-growing old trees from these pristine sites if there were no growth-longevity trade-off, but such trees are noticeably lacking from our graph for *Picea mariana* (ms Figure 1b).

In addition, the trade-offs are observed across many different species, including non-timber species and species from pristine tropical forests. In conclusion, we are confident that the mechanism (potential bias) suggested by the reviewer is not the driver for the observed trade-off.

Second, because tree-ring samples are often collected in extreme environments to support climate reconstructions, the fast-growing trees in better environments might be underrepresented in the tree-ring dataset. It is not clear to me how much of the dataset comes from the forest inventories and how much uses the existing tree-ring studies.

A sampling focus on extreme environments only results in a lack of fast growing, old individuals if the dendrochronologists also systematically disregard samples from old, fast-growing trees. Just focussing on marginal sites only results in general underrepresentation of fast-growing trees, but no specific lack of either old or young trees. Furthermore, there is not much evidence that old, fast growing trees were excluded from the ITRDB. On the contrary one would instead expect dendrochronologists to disregard young trees, as they are of no use for long-term climate reconstructions. Finally, we find the strongest trade-offs for the 12 species of the dataset from Quebec, that has been collected according to strict forest inventory rules from sites set out in a strict grid with no such bias for extreme environments.

Third, large old trees are often decayed in the center, making the determination of age impossible. I presume that such trees were eliminated from the current study, as they would be impossible to age, but we are not told. It is conceivable that old trees that grew fast in their youth are more likely to be rotten in the center because the conditions for decay organisms are also improved on better sites. Besides, the rapidly produced wood at the center might be less well defended than when it is produced more slowly. If they were more prone to rot, that would drop them out of this sample and contribute to the shape of the data in Fig. 1a and 1b. I ask that the authors tell us how the rotten cores were treated in the sampling and how that might influence the results.

Decaying centre wood affects both growth AND age estimates. In most species, missing the centre wood would lead to underestimations of ring width as it declines in most species as trees get older. Thus, missing the pith of trees (or cores with pith offsets) would likely bias our results in the opposite directions to what we observe, and cause (apparent) slow-growing trees to have lower ring counts.

However, if fast growing trees were more susceptible to rot, then we may indeed underestimate the age of fast-growing trees. It is hard to assess this bias for ITRDB-data as we do not know to what degree data are complete. However, for the Quebec dataset and those data contributed by co-authors (see ED Table 1), we have excluded all trees with missing tree ring sections. For the dataset from Quebec this was done by comparing cumulative ring widths to field diameters, and excluding any trees where the two measures deviated by more than 10%. As trade-offs in the Quebec dataset are stronger than those in the ITRDB, we are confident that the phenomenon mentioned by the reviewer does not explain the observed relationship.

We discuss these effects in our section "*Effects of pith offsets on early-growth age relationships*". Here we point out that wood rot may affect results, that growth generally declines with age, and we compare estimates of the trade-offs for Quebec and the ITRDB. WE have added a reference regarding the decrease of growth with tree age.

I was interested in the way they have estimated mortality due to increasing growth rates, and am particularly curious about how well they have described self-thinning. It has been noted that self-thinning causes mortality to increase as growth rates have increased (Pretzsch et al., 2014), but that stands still follow the old self-thinning lines. I would like to see how well the new mortality

calculations reproduce the self-thinning line. A demonstration of similar patterns would make the analysis more convincing. I understand that they have described maximum ages of survivors, which might be considered independent of self-thinning—except that the growth of survivors was influenced by the self-thinning during their lives—including perhaps their early lives. In any case, I think they should explain this more clearly.

We have to clarify here that the observed mortality increase (Fig. 3d) is a population-level increase in mortality for trees larger than 91mm in diameter. Our mortality function and our simulation approach is thus not suitable for addressing the questions the reviewer is asking here. We agree these questions are highly relevant and interesting to explore, but one would need an individual based model that includes mechanistic mortality functions for small-sized trees (< 91 mm) along with a mechanism allowing for effects of competition. These features were omitted from our stochastic simulations on purpose as we intended to develop a purely data driven simulator that would allow us assessing the effect of the observed trade-off on mortality and biomass stocks. In lines 199-203, we briefly discuss the consequences of this simplification on our core results “... , we did not simulate any competition effects, or changes in tree recruitment. One could argue that changes in climate or CO₂ increase will affect recruitment, and increases in standing biomass stocks will increase competition effects, leading to increased self-thinning and even stronger increases in mortality rates.”

And while they're at it, I would like to see some mention of which parameter is most valuable for land surface modeling: average longevity or maximum?

We do not know, as the two parameters are closely related (i.e., reduction in maximum longevity will also reduce the average). Our modelling exercise shows that reductions in life-span due to increased growth, results in a reduction of the average longevity of large trees (fig. 3e). If land surface models similarly include realistic population dynamics and mortality then the growth-mortality trade-off should be a prediction of the model. What is however more important -in our view- is that we work towards understanding the mechanisms behind the trade-off so that it can be incorporated mechanistically in the models.

Exceptions to the size-longevity tradeoff might also be worth considering.

We tried to find patterns in our dataset and identify which type of species were not showing the (negative) relationships seen across most others. However, no clear pattern emerges and more follow-up analysis is needed here.

I would start with the redwoods of northern California and the ancient Douglas-fir forests of west coast of North America (Waring & Franklin, 1979). Thanks to the late arrival of industrial forestry and the efforts of conservationists, some of these commercially valuable forests were preserved before they could be logged. Here one finds many of the largest, and perhaps oldest, individuals of several conifer genera in the world. I doubt that these individuals grew slowly when they were young, but it would be especially interesting to consider this case, at the other end of the size and growth-rate continuum from black spruce. There must be some such trees in British Columbia in the Canadian Forest Inventory. Perhaps they could be extracted from the mass of data to see if they fit the same patterns.

Unfortunately, very little public data are available on redwoods, but we do have a large number of Douglas-Fir sites and trees (147 sites, 2970 trees), many from the west coast, and the full set for this species shows a robust negative relationship between growth and longevity consistent with the overall pattern. For the Canadian Forestry Inventory, we only had access to the data from Quebec, and not from British Columbia.

These suggestions are all useful and these type of approaches of looking for species without trade-offs may help explaining the drivers. However, for the current manuscript we cannot include more analysis. Future analysis will focus on understanding where these trade-off are coming from.

The conclusion about neutralizing the carbon sink ignores forest management, which captures much of the forest production occurring in some parts of the world (Luyssaert et al., 2018). Sustainable forest harvesting could be accelerated if faster growth occurs, capturing the mortality before the mortality occurs. This would, of course, prevent the forests from reaching the age of natural mortality and would lead to a different set of fates for the carbon, including storage in wood products. In fact, this seems like a logical response to the patterns described in the current manuscript. I do not think this is a quibble if the authors wish to improve the parameterization of land surface models--or if they are interested in reaching a broad readership.

We had considered discussing these implications, but did not include them due to space limitations. We would like to keep the manuscript focussed on the effects of the trade-off for natural forest dynamics, and thus think that this aspect is out of scope for this manuscript.

Specific comments (overlapping with the above):

Line 77: Do the global models need the maximum lifespans or the averages? Almost all of tree mortality occurs due to self-thinning or forest management. How relevant are these maxima globally?

See comments re self-thinning above.

Lines 106-131: These are not so much causes as correlations, by my reckoning. As an ecophysiologicalist, I would like to know what physiological process is disrupted uniformly across this range of sites. This is analogous to the cavitation or carbon starvation question...or maybe it's exactly the same question. In any case, this manuscript would be strengthened by a tie to the ongoing development of mechanistic thinking about tree mortality. Should cite Koch et al. about the physiological basis of max ht (Koch et al., 2004) and the influence of height on drought-induced mortality (Stovall et al., 2019)

We fully acknowledge that we have -in fact- not been able to pinpoint a true mechanistic basis for the trade-off. To reflect this, we have changed the title of the section to "Environmental controls of the trade-off", and deleted the first sentence.

We added the two suggested references to the manuscript in the section where we mention the hydraulic limitation theory (lines 162-165: *what ultimately kills a tree once it exceeds its maximum potential size may involve hydraulic limitation (Ryan and Yoder 1997, Koch et al. 2004, Stoval et al. 2019)*).

149-168: tree size height and diameter are intermingled in a way that is confusing here because height limitations might occur due to hydraulics, but allocation to diameter growth might well be favored by height limitations. So there's a question underneath this discussion about size related to allometric shifts with height.

We deliberately avoided this discussion because we know very little about the mentioned allocational shifts, and because they may be highly species-specific. However, as a general rule, one still expects that the highest trees are generally also the 'fattest' trees and thus one would expect mortality to increase with diameter, even if such allocational shifts were to occur.

210: But Pretzsch(Pretzsch et al., 2014) argues that so far they haven't. Should say so.

We have changed this, and now say, "*A more likely scenario that could potentially account for greater future forest carbon storage under rising CO₂ is that tree size-density relationships could be modified (Kubiske et al. 2018), although long-term empirical data show that self-thinning rules remained constant despite strong growth increases over time (Pretzsch et al.2014).*"

210-212: Right, but there will be significant mortality while the shifts are taking place. Is that within the time frame of these models or are they steady-state?

This is not captured by our models as we model growth in a fixed latitude for a fixed species in response to temperature increases. What we refer to are possible poleward shifts in species distribution (and general forest cover) from south to north, which may allow for greater carbon storage. However, this is outside the scope of our modelling framework.

Fig. 2 errors? Data?

We think the reviewer wants us to add errors on the regression lines. We can do this but believe that adding errors to the relationships will make the patterns in the figure only harder to interpret for the reader. We have now however added to the legend that regressions are significant and we have changed non-significant relationships in the figure to stippled lines.

Fig 3a. two data points support the exponential increase at the end, but more do not. In fact, it is not clear to me that the exponential curve fits better than a straight line would. This could lead to a strong overestimate of mortality. It is stated in the text (line 439) that there was no bias in the exponential fits, but this figure seems to show otherwise.

Note that our statement on the bias of the exponential fit (in line 439) refers to the fit to assess the growth-longevity relationship and not the mortality relationship shown in figure 3a.

The mortality increase with diameter is indeed not supported by all points at larger diameter classes, and this is probably because of increased uncertainty due to lower sample sizes for larger trees.

However, the used mortality curve does a very good job at predicting the growth longevity trade-off (see Fig. 3b). As evaluating the effect of this trade-off was the main purpose of the simulation, we preferred to use this mortality curve above other alternatives that will result in weaker less realistic trade-offs.

Why ring widths and not basal area increase? The same amount of wood produces a narrower ring on a bigger tree. Does that matter here?

The difference between the use of ring width and basal area is small for small diameter trees, but indeed very large when trees are big. Thus, for the purpose of this study where we only looked at growth over the first ten years, the two measures would give very similar results. Some errors due to missing piths would be exacerbated by using basal area metrics, while ring width can be extracted directly from the data and compared with results from previous studies.

473-480: good

547: No. It shows that the biases you tested were not significant. See questions about other biases above.

See our response to the other biases. We have added text to the manuscript.

Koch, G. W., Sillett, S. C., Jennings, G. M., & Davis, S. D. (2004). The limits to tree height. *Nature*, 428(6985), 851–854.

Luyssaert, S., Marie, G., Valade, A., Chen, Y.-Y., Njakou Djomo, S., Ryder, J., Otto, J., Naudts, K., Lansø, A. S., Ghattas, J., & McGrath, M. J. (2018). Trade-offs in using European forests to meet climate objectives. *Nature*, 562(7726), 259–262. <https://doi.org/10.1038/s41586-018-0577-1>

Sevanto, S., McDowell, N. G., Dickman, L. T., Pangle, R., & Pockman, W. T. (2014). How do trees die? A test of the hydraulic failure and carbon starvation hypotheses. *Plant, Cell & Environment*, 37(1), 153–161. <https://doi.org/10.1111/pce.12141>

Stovall, A. E. L., Shugart, H., & Yang, X. (2019). Tree height explains mortality risk during an intense drought. *Nature Communications*, 10(1), 4385. <https://doi.org/10.1038/s41467-019-12380-6>

Waring, R. H., & Franklin, J. F. (1979). Evergreen Coniferous Forests of the Pacific Northwest. *Science*, 204(4400), 1380–1386. <https://doi.org/10.1126/science.204.4400.1380>

Reviewer #3 (Remarks to the Author):

Overall the paper presents convincing evidence of early growth rate -lifespan trade-offs globally, as well as possible explanations for these trade-offs.

The authors apply simple simulations that assume mortality rates that increase with size.

Note that we do not simply assume mortality but in fact derive it empirically from the Quebec-NFI data, and then assess the effect of the mortality curve on the trade-offs.

These produce growth-lifespan relationships similar to those that are observed, whereas an assumption of age-dependent mortality does not.

The authors conclude that growth-lifespan tradeoffs are missing from current Earth system models which must therefore be incorrectly predicting a continuation of the carbon sink into mature forests into the future.

We feel this is not entirely fair as it is not quite what we are saying. What we are demonstrating is (i) that there is very strong evidence of the existence of a growth-longevity trade-off and (ii) that the existence of such a trade-off strongly suggests that the effect of a tree growth stimulation on carbon stocks is a transient phenomenon. That does not even need a model - although our heavily data based and data driven -model does demonstrate the point very clearly.

My main concern with this paper is that the extrapolation of the findings to conclusions about the incorrectness of Earth system models is not well-founded. The reference (3) (Sitch et al. 2008) to models examines projections of Dynamic Global Vegetation Models (DGVMs) not Earth system models. and there has been much subsequent development in the DGVM community. Many DGVMs now include stand demography (including recruitment, competition and size-dependent mortality), from which may emerge growth-lifespan trade-offs. See Fisher et al. 2017, 10.1111/gcb.13910 for a more up to date and comprehensive review of the current generation of DGVMs.

It may indeed be that most recent Earth System Models may be able to reproduce the trade-off we highlight, although to our knowledge there is no publication documenting this. Also, there are only few ESMs which now include population dynamics. More traditional DGVM's and ESM's have certainly not had the capability to represent the trade-off documented in our manuscript and these have often been invoked to suggest that CO₂ fertilization will cause a substantial carbon sink.

To reflect better recent advances of ESM's, we now phrase more cautiously in the introduction and end of discussion. In the intro we have added a reference to Fisher et al. and say “... *current, incomplete knowledge of the universality and causes of the feedback hinders its representation in Earth System Models and thus is an important uncertainty in predictions of future forest carbon uptake in response to global change*^{4-6,8}. “ We believe this is a fair statement.

In the discussion we now more cautiously say that “*This mechanism is at odds with **most extant** Earth System Model simulations.* “

My second main concern is about the final recommendation (l 234) that growth-rate mortality trade-offs be integrated into process-based simulations of forest carbon storage. This is not an immediately helpful suggestion, as the process or processes from which these trade-offs emerge are not well understood- as discussed in the section headed “Causes of the trade-off”.

We agree that it is not an 'immediately helpful suggestion', and that the section header is misleading. We reworded the section header (also in response to reviewer 2). However, our statement, that if DGVMs are not able to reproduce the trade-off, they miss a first order constraint

and thus are unlikely to predict the carbon sink / source of the land vegetation, is correct. This is in our opinion important to be pointed out. It is this fact which we want to highlight.

A more constructive use of the findings could be to recommend how the data relating growth rate to life-span, or a synthesis thereof, could be used as a benchmark for DGVMs.

This is well taken but this is not the purpose nor the point of the paper. However, we agree that this is a good topic, e.g. for a PhD thesis, and thus for a paper in the future.

Minor comments:

Line 36: this comment on current Earth system models is not well supported by the article.

Here, we have to disagree with the reviewer as the existence of the trade-off identified across many taxa, strongly suggests that the effect of a tree growth stimulation on carbon stocks is a transient phenomenon. Our simulations back this up.

Line 41: generality here is misleading: increases in temperature don't always lead to growth stimulation.... In warm climates, sensitivity of growth to T may be negative.

True and we have refined the statement to *"increases in CO₂ and temperature, in cold regions, are often believed to have stimulated tree growth."*

Line 45: reference (3) is insufficient to support this: the models assessed by Sitch et al. are not the same as in Earth system models.

We agree and now also cite Cox et al. 2019.

Cox, P. M. Emergent constraints on climate-carbon cycle feedbacks. *Current Climate Change Reports* 5, 275-281 (2019)

Reviewers' comments second round:

Reviewer #2 (Remarks to the Author):

In my earlier review (R2) of this manuscript, I asked the following:

Fig 3a. two data points support the exponential increase at the end, but more do not. In fact, it is not clear to me that the exponential curve fits better than a straight line would. This could lead to a strong overestimate of mortality. It is stated in the text (line 439) that there was no bias in the exponential fits, but this figure seems to show otherwise.

This was the response:

Note that our statement on the bias of the exponential fit (in line 439) refers to the fit to assess the growth-longevity relationship and not the mortality relationship shown in figure 3a. The mortality increase with diameter is indeed not supported by all points at larger diameter classes, and this is probably because of increased uncertainty due to lower sample sizes for larger trees. However, the used mortality curve does a very good job at predicting the growth longevity trade-off (see Fig. 3b). As evaluating the effect of this trade-off was the main purpose of the simulation, we preferred to use this mortality curve above other alternatives that will result in weaker less realistic trade-offs.

This led me to wonder if the simulated high mortality rates were an artefact of the process used to fit them. So I started trying to fit the data in Fig. 3a to see if I could reconstruct equation 4. I could not. If I put a 200 mm tree into the equation, the mortality rate = $0.003 \cdot (200 - 91)^4$, or 423474. Fig. 3a shows values between 0 and 0.25. This seems too simple for a math error. This estimate of mortality is critical in the modelling and it needs to be fixed.

More broadly, it is necessary to describe, at least briefly, how the model parameters were obtained. The fact that this parameterization made the relationship in 3b look good is not sufficient. Nor is the authors' opinion that a weaker trade-off is necessarily more "realistic." This logic is circular.

Perhaps equation 4 will be an easy fix, but I don't know how to proceed until this problem is corrected. Some objective means of fitting it seems like a fair request given its central role in the comparison of the models.

John Marshall

REVIEWER COMMENTS

Reviewer #2 (Remarks to the Author):

In my earlier review (R2) of this manuscript, I asked the following:

Fig 3a. two data points support the exponential increase at the end, but more do not. In fact, it is not clear to me that the exponential curve fits better than a straight line would. This could lead to a strong overestimate of mortality. It is stated in the text (line 439) that there was no bias in the exponential fits, but this figure seems to show otherwise.

This was the response:

Note that our statement on the bias of the exponential fit (in line 439) refers to the fit to assess the growth-longevity relationship and not the mortality relationship shown in figure 3a. The mortality increase with diameter is indeed not supported by all points at larger diameter classes, and this is probably because of increased uncertainty due to lower sample sizes for larger trees. However, the used mortality curve does a very good job at predicting the growth longevity trade-off (see Fig. 3b). As evaluating the effect of this trade-off was the main purpose of the simulation, we preferred to use this mortality curve above other alternatives that will result in weaker less realistic trade-offs.

This led me to wonder if the simulated high mortality rates were an artefact of the process used to fit them. So I started trying to fit the data in Fig. 3a to see if I could reconstruct equation 4. I could not. If I put a 200 mm tree into the equation, the mortality rate = $0.003 \cdot (200 - 91)^4$, or 423474. Fig. 3a shows values between 0 and 0.25. This seems too simple for a math error. This estimate of mortality is critical in the modelling and it needs to be fixed.

More broadly, it is necessary to describe, at least briefly, how the model parameters were obtained. The fact that this parameterization made the relationship in 3b look good is not sufficient. Nor is the authors' opinion that a weaker trade-off is necessarily more "realistic." This logic is circular.

Perhaps equation 4 will be an easy fix, but I don't know how to proceed until this problem is corrected. Some objective means of fitting it seems like a fair request given its central role in the comparison of the models.

John Marshall

Response to reviewer's comments

We would like to thank the reviewer for pointing out the mistake made in equation 4 of the manuscript. This is a typo and is not what we used for the simulations. In the previous manuscript we got the decimals wrong for k and missed a constant b in the formula. The correct equation is:

$$\mu(D) = \begin{cases} 0 & \text{for } D \leq D_0 \\ b + k \cdot (D - D_0)^4 & \text{for } D > D_0 \end{cases}$$

with $b=0.025$, $k=3 \cdot 10^{-11}$ and $D_0=91$ mm. We corrected this formula in the manuscript.

The reviewer asks how we obtained the model parameters and states that the fit to the mortality model is not good enough. We agree with the reviewer that the fit is not optimal, but the mortality model was not a formal fit. The model was chosen to reproduce the growth longevity trade-off (see Fig. 3b).

We now assessed the sensitivity of the simulations to variations in size dependent-mortality. To this end, we fitted a linear model to the mortality data, leaving out those two outliers with mortality rates $> 0.1 \text{ \% yr}^{-1}$. The results of these simulations with different mortality functions are shown in the response figure below.

This analysis provides two important insights.

1. Even with linearly increasing mortality rates (panel a, red line) we find a close resemblance between the simulated trade-off and the observed trade-off (panel b). Thus, the strength of the simulated trade-off is robust with regard to the choice of mortality function.
2. The change of the mortality function has only a small effect on the observed dynamics. By changing the mortality function to a linear one, we find that trees live slightly less long (panel e, red line), due to the slightly higher mortality in mid-diameter classes 150-275 mm (panel a). This faster turnover results in the simulations in an even faster return of standing biomass stocks to original levels after a growth stimulation, as compared to the simulation with a fourth power mortality increase (panel f).

In all, this shows that the main conclusions of our simulations are robust with regard to the use of mortality function.

We have now clarified in the manuscript that the mortality curve is not a formal fit but a chosen model to represent the trade-off, and we added a statement that the simulation outcome is not sensitive to the shape of the mortality curve (lines 493-496).

Response Figure. Red lines indicate the new simulation outcome using the linear mortality function from panel a. Panel b compares simulated trade-off to the observed. For further information compare to ms main figure 3.

Reviewers' comments third round:

Reviewer #2 (Remarks to the Author):

This is a fair and reasonable response to my earlier concerns. This is an interesting paper and I am glad for having the chance to "discuss" it with the authors.

John Marshall